# Impact of Paraffin Composition on the Interactions between Waxes, Asphaltenes, and Paraffin Inhibitors in a Light Crude Oil

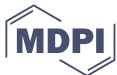

Oualid M'barki [1], John Clements [2], Luis Salazar [2], James Machac, Jr. [2] and Quoc P. Nguyen [1,*]

1   Hildebrand Department of Petroleum and Geosystems Engineering, The University of Texas at Austin, 200 E. Dean Keeton Stop C0300, Austin, TX 78712-1585, USA
2   Indorama Ventures, 24 Waterway, The Woodlands, TX 77380, USA
*   Correspondence: quoc_p_nguyen@mail.utexas.edu; Tel.: +1-512-471-1204

**Abstract:** The effect of wax molecular weight distribution on the efficacy of two alpha olefin-maleic anhydride paraffin inhibitors (PIs) having different densities of alkyl side-chains were examined in light West Texas crude in the absence and presence of asphaltenes. Interpretation of the data was aided by cross-polarization microscopy. Primary differences in wax crystal morphology appear to be driven by the composition of the wax, with secondary differences being associated with the choice of PI. In the absence of asphaltenes, the effect of wax composition on PI performance (i.e., reducing oil viscosity and wax appearance temperature) is greater for the PI having the higher chain density, with the one having the lower chain density being generally more effective regardless of the wax composition. These differences are diminished in the presence of asphaltenes such that the PI having the higher chain density is somewhat more effective. Trends in both morphology and viscosity suggest a steric effect associated with wax composition that is lessened on interaction of the PIs with asphaltenes.

**Keywords:** paraffin composition; wax appearance temperature (WAT); asphaltenes-wax interactions; alpha olefin-maleic anhydride polymers





## 1. Introduction

The detrimental effects of wax formation on flow assurance in the oil and gas industry have been extensively elaborated in the literature [1–5]. Paraffin inhibitors (PIs) are often used to reduce the oil pour point and wax appearance temperature (WAT) to mitigate wax precipitation and deposition in flowlines. Comb polymers with non-polar alkyl chains attached to polar backbones have proved to interact with large paraffin molecules via nucleation, adsorption, and co-crystallization that inhibits wax crystal growth as well as modifies crystal morphology for reduced oil viscosity and WAT [6,7]. The side-chain length of comb shaped PIs can strongly affect the wax inhibition efficacy [8–10]. Tailoring the structural properties of the PI, such as alkyl chain length distribution, to enhance its interaction with a specific composition of wax and of crude oil could be critical to PI efficacy [11]. Indeed, our previous study of the impact of PI structure on wax morphology and associated oil rheology in a simple dodecane system showed that the most important factor to improve PI performance is the functionality associated with the point of attachment of pendant alkyl chains to the backbone of the comb polymer such as ester linkage [12]. Particularly, the robustness of a paraffin inhibition polymer in controlling waxy oil viscosity appears to be more sensitive to side-chains density rather than side-chains length, which is remarkable as matching the length of paraffin carbon chain with the alkyl side chain of the comb PI has been the most common method for PI design [13]. Another striking observation was the ability of asphaltenes to reduce WAT in dodecane and the synergy between wax, PI and asphaltenes to further reduce WAT and oil viscosity at low temperature, advancing our understanding of the functionality of synthetic PIs and asphaltenes as a PI [14–20].

The present work continues to advance the understanding of waxy oil behavior in the presence of alpha olefin-co-maleic anhydride (AO-MA) paraffin inhibitors, with a focus on the impact of wax composition, particular chain length, i.e., molecular weight, and, to a lesser extent, the breadth of the distribution, in a more complex oil system by substituting the dodecane by a light crude oil, referred as West Texas crude (WT). Three wax compositions were selected, having different average chain lengths. The interaction between asphaltenes with these waxes and its impact on the efficacy of PIs in modifying paraffin crystal morphology and rheological behavior of crude oil are also investigated with the help of cross polarization microscopy. A detailed description of the materials and experimental procedures is given in the following section.

## 2. Materials and Methods

### 2.1. Materials

Three different wax blends, *CW1*, *CW2*, and *CW3*, having different average molecular weights and molecular weight distributions were prepared by blending three different commercial waxes, *CW1*, *CW2*, and *CW3* (Table 1), obtained from Sigma–Aldrich. The carbon number, i.e., chain length, distributions of each component are illustrated in Figure 1, ranges primarily from C18 to C38 as these lengths tend to form macrocrystalline structures that lead to wax precipitation and gelation issues. *CW1—3* are distributions of light, medium and heavy paraffins with average carbon numbers and standard deviations of 24.2 ± 3.3, 26.6 ± 2.3, and 29.1 ± 3.8, respectively.

**Table 1.** Wax composition and melting ranges.

| Wax Blend | Commercial Wax | | | | | | Properties (Carbon No.) | |
| --- | --- | --- | --- | --- | --- | --- | --- | --- |
| | CW1 | | CW2 | | CW3 | | | |
| | Wt. % | MR (°C) | Wt. % | MR (°C) | Wt. % | MR (°C) | Avg. | Std. Dev. |
| W1 | 45 | | 55 | | — | | 24.2 | 3.3 |
| W2 | — | 42–44 | 100 | 53–58 | — | 65+ | 26.6 | 2.3 |
| W3 | — | | 55 | | 45 | | 29.1 | 3.8 |

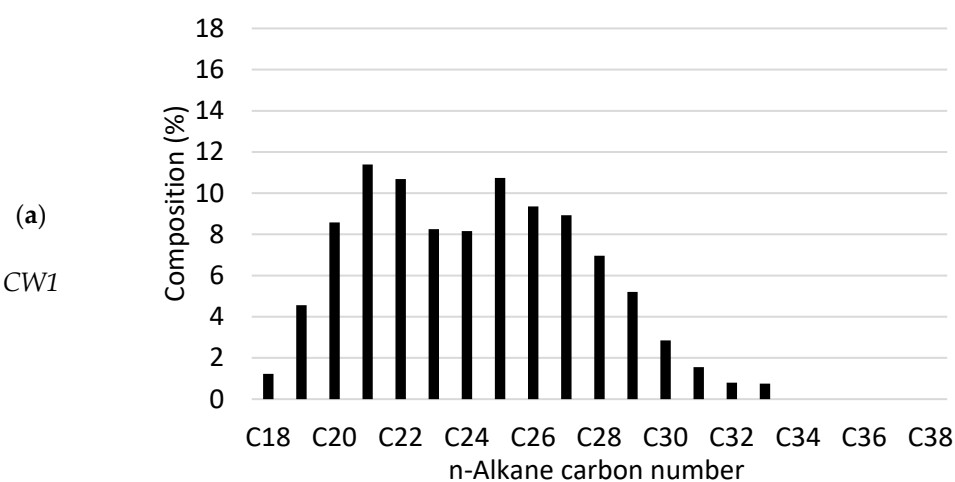

(**a**)

*CW1*

**Figure 1.** *Cont.*

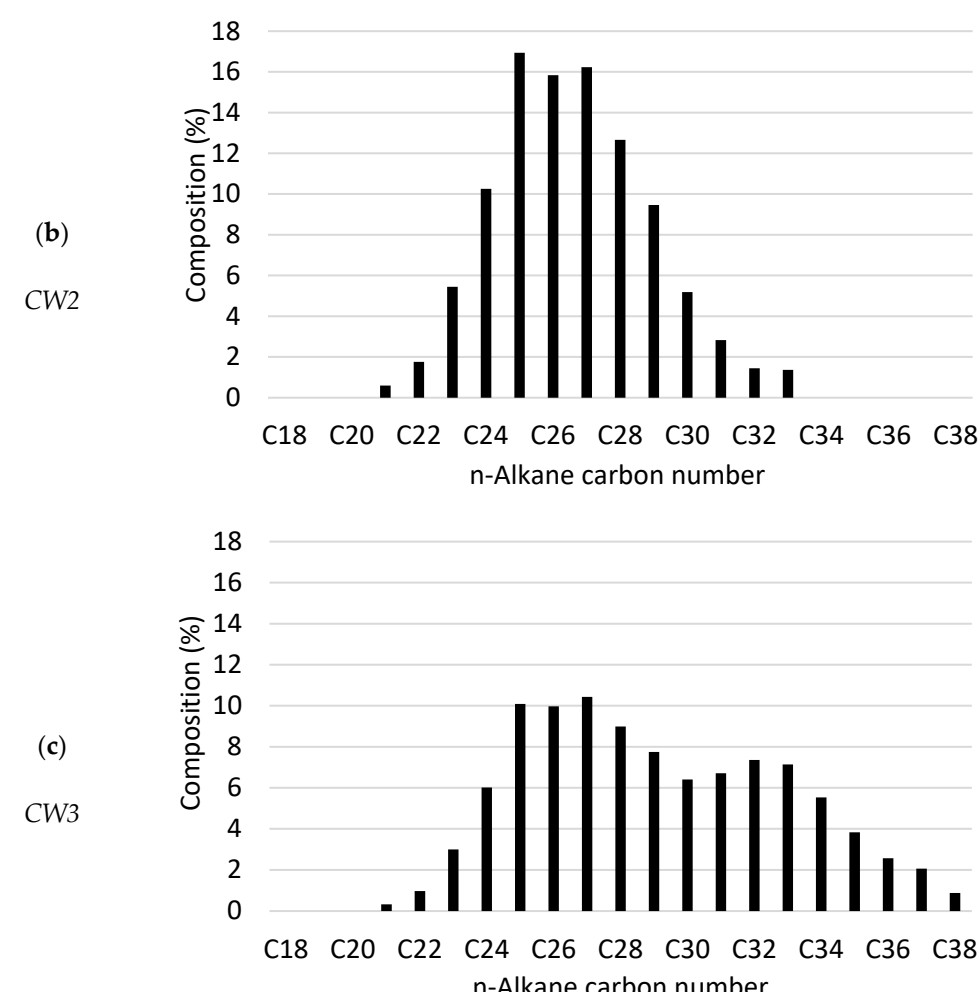

**Figure 1.** n-Alkane carbon number distribution of wax blends: (**a**) *CW1*, (**b**) *CW2*, and (**c**) *CW3* obtained by gas chromatography.

PI2 and PI3 are commercially available PIs provided by Indorama Ventures Oxides and Derivatives LLC. These PIs possess AO-MA alternating copolymer backbones and saturated, hydrocarbon tines with a weight-average molecular weight of ~5000 g·mol$^{-1}$ (GPC, THF, 1 mL/min, poly(propylene glycol) standards). Side-chains are attached via reaction of the anhydride in the AO-MA copolymer with mixtures of fatty alcohols to give ester linkages, with the molar ratio of maleic anhydride to alcohol, hereafter referred to as the *chain density*, being 1.6 in PI2 and 1.2 in PI3 (Table 2). The distribution of hydrocarbon chains attached via the above-mentioned linkages as well as those originating from the alpha olefin monomers themselves vary in length from C16 to C26. Both PIs contain similar mixtures of petroleum-derived heavy aromatic naphtha solvents and have similar polymeric concentrations of about 82 wt. %. A more detailed description of these paraffin inhibitors can be found in [12].

**Table 2.** Paraffin inhibitor (PI) composition and properties.

| PI | Side-Chain Carbon No. Distribution | | |
|---|---|---|---|
| | Range [1] | Avg. [2] | Density [3] |
| PI2 | 18–26 | 21.1 ± 2.2 | 1.6 |
| PI3 | 18–22 | 20.7 ± 1.7 | 1.2 |

[1] Breadth of carbon number distribution, including all lengths > 2% of the most abundance length. [2] Average value ± standard deviation. [3] Number of alkyl chains attached through each anhydride.

### 2.2. Methods

### 2.2.1. Solution Preparation

Wax-containing oil samples were prepared by diluting one of the three wax compositions *CW1*, *CW2*, *CW3* to 10 wt. % in WT light crude oil. This dead crude has a density of 849 kg/m$^3$ at standard conditions and does not exhibit a WAT over the temperature range from 8 to 50 °C. The asphaltene (heptane insolubles) content of the crude was not detectable by ASTM D2007-80. Oil samples containing 500 ppm of the polymeric component of each PI and, in some cases, 1 wt. % asphaltenes were prepared similarly (Table 3). Further, 1 wt. % asphaltenes were prepared by mixing a predetermined amount of an asphaltic crude sample that contents 20 wt. % asphaltenes (heptane insolubles) into the WT light crude oil using an ultrasonic homogenizer. Once prepared, the oil samples were heated to 75 °C to completely solubilize the wax, then characterized by cross-polarized microscopy (CPM) and rheology.

**Table 3.** Synthetic oil samples, each containing 10 wt. % wax.

| Oil | Wax | PI | Oil | Wax | PI |
|-----|-----|-----|-----|-----|-----|
| No Asphaltenes | | | 1 wt. % Asphaltenes | | |
| 1 | | — | 1A | | — |
| 1-PI2 | *CW1* | PI2 | 1A-PI2 | *CW1* | PI2 |
| 1-PI3 | | PI3 | 1A-PI3 | | PI3 |
| 2 | | — | 2A | | — |
| 2-PI2 | *CW2* | PI2 | 2A-PI2 | *CW2* | PI2 |
| 2-PI3 | | PI3 | 2A-PI3 | | PI3 |
| 3 | | — | 3A | | — |
| 3-PI2 | *CW3* | PI2 | 3A-PI2 | *CW3* | PI2 |
| 3-PI3 | | PI3 | 3A-PI2 | | PI3 |

### 2.2.2. Cross-Polarized Microscopy (CPM)

Oil samples were equilibrated at 20 °C for one hour prior to CPM analysis to ensure that wax crystals were fully developed and stable in the solution. An oil drop taken from each sample was placed between two glass plates and images of wax in the drop were recorded using a BX51 cross-polarized microscope (Olympus) equipped with a 40× magnification camera and a nitrogen driven heating and cooling stage to regulate temperature. CPM images of transient waxy oil behavior on cooling from high to low temperature could not be obtained because the heating/cooling system available was equipped with manual rather than automatic temperature control.

### 2.2.3. Viscosity Measurement

An Ares LS-1 rheometer (TA instruments) equipped with a Couette geometry was used to measure the viscosity of oil samples. Cup and bob diameters were 30 and 28 mm. Oil viscosity over a temperature range from 8 to 50 °C was measured at a constant shear rate of 25 s$^{-1}$. During these measurements, the sample was cooled from 50 to 8 °C and warmed back to 50 °C at a constant cooling/heating rate of 1 °C/min. Experiments were performed in duplicate to verify reproducibility.

WAT values for each sample were obtained from their viscosity curves as the onset temperature at which the rate of viscosity increase becomes more dramatic with further cooling [12]. The shear rate of 25 s$^{-1}$ was chosen because it has been commonly used in previous laboratory characterizations of waxy oil rheology and closely represents boundary layer flow over borehole wall or tubbing/casing wetting surfaces. Previous studies of the effect of shear rate on waxy oil rheology showed that non-Newtonian (i.e., shear thinning) behavior could become significant at low shear rates and high wax content [12]. While the main focus of this work is the impact of wax composition on the interactions between

asphaltenes and paraffin inhibitors at a given shear rate, the results may vary with shear flow conditions (shear stress and shear rate) in hydrocarbon wells.

## 3. Results and Discussion

Selection of comb copolymer PIs often involves attempts to match the wax composition in the oil to be treated with the length distribution of side-chains in the PI. In this work, two inhibitors that differ somewhat in the length and breadth of their side-chains yet notably in their chain densities were selected for study as the latter structural feature appeared to have a greater effect on PI performance over a larger dataset evaluated in dodecane-based synthetic oils [12]. The data reported herein is interpreted in three ways. Firstly, the effect of wax chain length on crystal morphology and viscosity is examined, with the former data being employed to interpret the latter. Secondly, the performance of PI2 and PI3 are compared with emphasis on differences arising from wax chain length. Lastly, the effect of asphaltenes is discussed.

### 3.1. Rheological Characterization and Microstructure of Oil Containing Wax

CPM images of oils 2 and 3 following equilibration at 20 °C are shown in Figure 2. Wax crystals were not clearly observed in oil 1 even though the WAT is 7.5 °C above the equilibration temperature, possibly due to the tendency of this wax, having a shorter average carbon number, to form smaller crystals and weaker networks. This is consistent with the measured viscosities of these oils, i.e., the viscosity of oil 1 is about 300 cP at 20 °C whereas, that of oils 2 and 3 is about 800 cP. Although slight differences in morphology are present, the wax crystal networks arising from wax compositions *CW2* and *CW3* do not appear notably dissimilar, possibly because their viscosities at 20 °C are nearly equal.

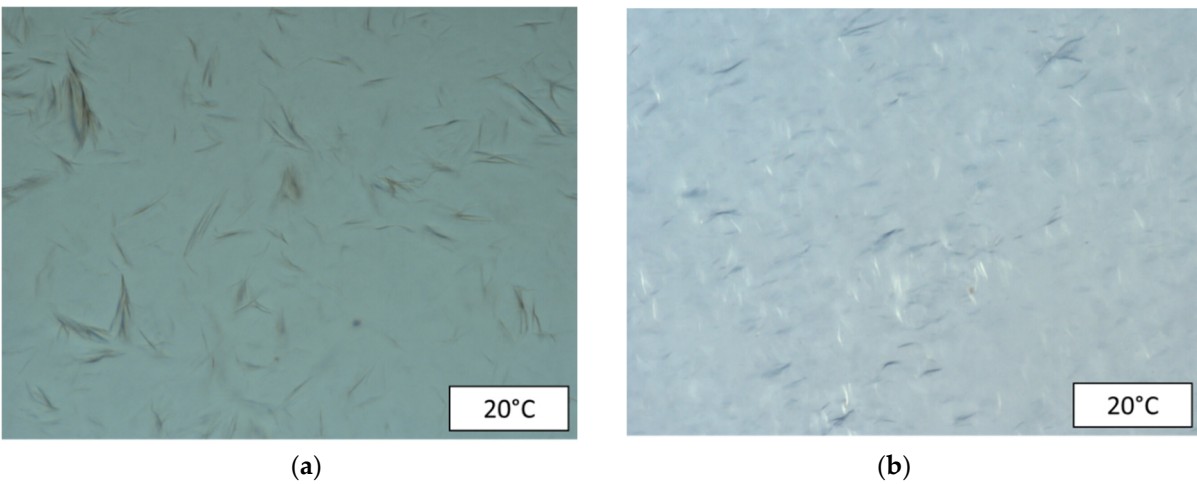

                        (**a**)                                          (**b**)

**Figure 2.** CPM images at 20 °C of oils (**a**) 2 and (**b**) 3. Wax crystals were not observed for oil 1 under the same conditions.

The viscosities of oils 1, 2, and 3, containing 10 wt. % of wax compositions *CW1*, *CW2*, and *CW3*, respectively, as a function of temperature over the cooling/heating cycle 50 °C → 8 °C → 50 °C are shown in Figure 3. The WAT values determined from the viscosity curves shown in Figure 3 are 27.5, 30 and 38.5 °C for oils 1, 2, and 3, respectively. The WAT appears to increase with the average paraffin chain length. Their viscosities continue to increase on cooling below their WATs, reaching maximum values of 834, 788, and 991 cP at 9 °C. Table 4 presents a summary of WATs, maximum viscosity and the viscosity reduction observed around 9 °C. As oil 1 is cooled below the WAT, its viscosity increases sharply to 311 cP at 24 °C, then decreases to 216 cP at 22.5 °C before increasing again with further cooling. Interestingly, this feature, which may be attributed to wax crystal relaxation and disentanglement, was not observed for oils 2 or 3 and is somewhat counter-intuitively confined to the wax shorter chain composition *CW1*. Significant hysteresis

between the cooling and heating cycles were observed due to the energy required to form and subsequently disrupt wax crystal networks, with larger hysteresis being observed for oils 2 and 3 relative to oil 1. Additionally, the oil viscosities on re-heating to 50 °C are slightly greater by about 1 cP than their pre-cooled values, likely owing to incomplete dissolution of the crystal network, which can be a somewhat slow process.

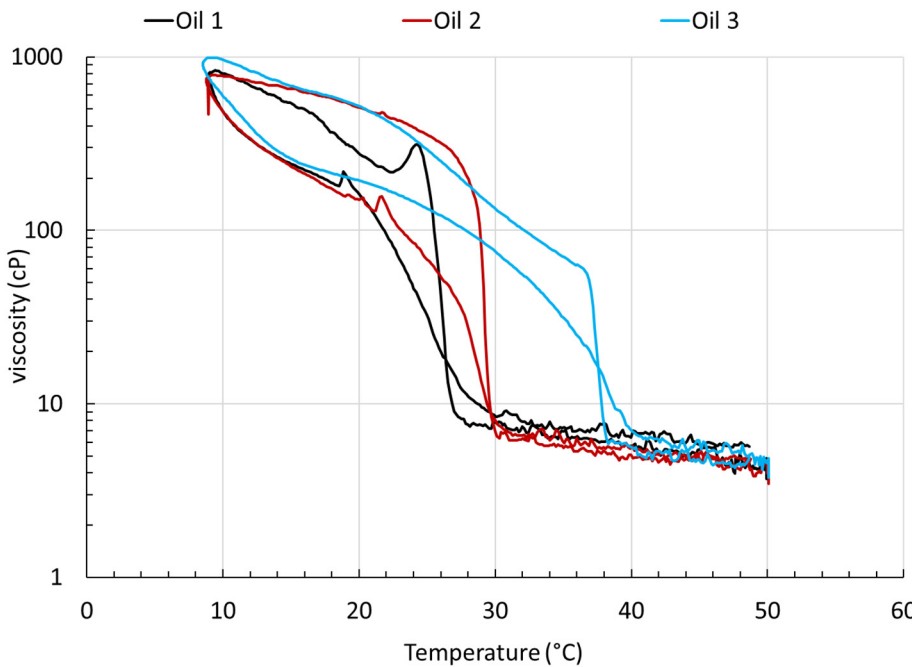

**Figure 3.** Viscosity as a function of the cooling/heating cycle 50 °C $\rightarrow$ 8 °C $\rightarrow$ 50 °C for oils 1, 2, and 3, measured by applying a constant shear of 25 s$^{-1}$ and a cooling/heating rate of 1 °C/min.

**Table 4.** WATs, maximum viscosity reached and percent viscosity reduction of oils relative to their untreated counterpart oils 1, 2 and 3. Negative values indicate viscosity increase.

| Oil | WAT (°C) | $\mu_{max}$ (cP) | Reduction (%) | | WAT (°C) | $\mu_{max}$ (cP) | Reduction (%) |
|---|---|---|---|---|---|---|---|
| | | | Wax composition *CW1* | | | | |
| Oil 1 | 27.5 | 834 | - | Oil 1A | 25 | 655 | 21.5 |
| Oil 1-PI2 | 26 | 28 | 96.6 | Oil 1A-PI2 | not defined | 36 | 95.7 |
| Oil 1-PI3 | 17 | 202 | 75.8 | Oil 1A-PI3 | not defined | 202 | 75.8 |
| | | | Wax composition *CW2* | | | | |
| Oil 2 | 30 | 788 | - | Oil 2A | 30 | 900 | −14.2 |
| Oil 2-PI2 | 28 | 204 | 74.1 | Oil 2A-PI2 | not defined | 117 | 85.2 |
| Oil 2-PI3 | 28 | 337 | 57.2 | Oil 2A-PI3 | not defined | 100 | 87.3 |
| | | | Wax composition *CW3* | | | | |
| Oil 3 | 38.5 | 991 | - | Oil 3A | 36 | 1010 | −1.9 |
| Oil 3-PI2 | 36 | 327 | 67.0 | Oil 3A-PI2 | not defined | 37 | 96.3 |
| Oil 3-PI3 | 17 | 458 | 53.8 | Oil 3A-PI3 | not defined | 80 | 91.9 |

### 3.2. Rheological Characterization and Microstructure of Oil Containing Wax, Treated with PI

No crystals were observed in the CPM images of oils 1-PI2 or 1-PI3, each containing the lighter wax composition *CW1*. The set of oils 2-PI2 and 2-PI3 (Figure 4a,c), each containing wax composition *CW2*, are similar in their primary morphologies, as are the set of oils 3-PI2 and 3-PI3 (Figure 4b,d), each containing composition *CW3*. However, the morphologies found in each set are quite different from one another. As such, primary differences in morphology appear to be driven by the wax composition rather than the selection of PI in

these cases, where the latter selection eventuates in less prominent, secondary differences in morphology. In the presence of either PI, more ordered, needle-like spherulites were observed in oils containing the heavier wax composition *CW3* than those containing *CW2*. Although a correlation between the chain density of the PI and crystal spherulite size was observed previously in a dodecane-based model, no such correlation is discernable in this work.

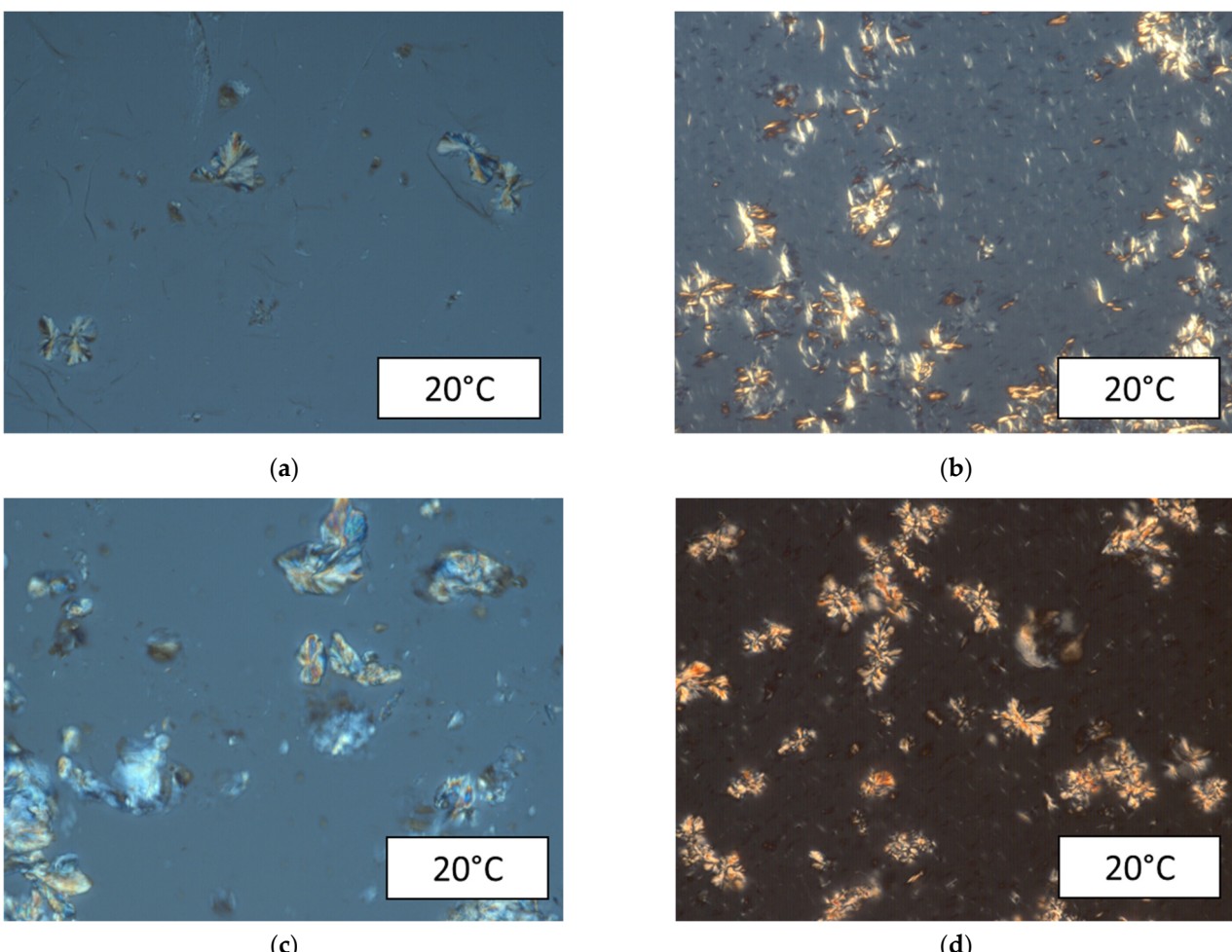

**Figure 4.** CPM images at 20 °C of oils (**a**) 2-PI2, (**b**) 3-PI2, (**c**) 2-PI3, and (**d**) 3-PI3. Wax crystals were not observed for oils 1-PI2 or 1-PI3 under the same conditions.

The viscosities of oils 1, 2, and 3, each treated with 500 ppm of the polymeric component of PI2, 1-PI2, 2-PI2, and 3-PI2, as a function of temperature over the cooling/heating cycle 50 °C → 8 °C → 50 °C are shown in Figure 5a. Likewise, the profiles of the analogous oils treated with PI3, 1-PI3, 2-PI3, and 3-PI3, are shown in Figure 5b. Oils 1-PI2, 2-PI2, and 3-PI2, each treated with PI2 and containing medium and heavy paraffin compositions *CW1*, *CW2*, and *CW3*, respectively, each exhibit only a modest 1.0–1.5 °C reduction of WAT relative to their untreated controls. Of perhaps greater significance is the overall reduction of the viscosity profiles, with maximum viscosities realized on cooling to about 9 °C reduced to 28, 204 and 327 cP for oils 1-PI2, 2-PI2, and 3-PI2, respectively (Table 4). Oils 1-PI3 and 3-PI3, each treated with PI3, show marked reductions in their WATs to about 14 and 21 °C, with that of oil 2-PI3 being more gradual over the range 20–27 °C. The maximum viscosities of these oils, also realized on cooling to about 9 °C, are reduced to 202, 337, and 458 cP for oils 1-PI3, 2-PI3, and 3-PI3, respectively. Unsurprisingly, the effectiveness of both PIs follows the trend *CW1 > CW2 > CW3*.

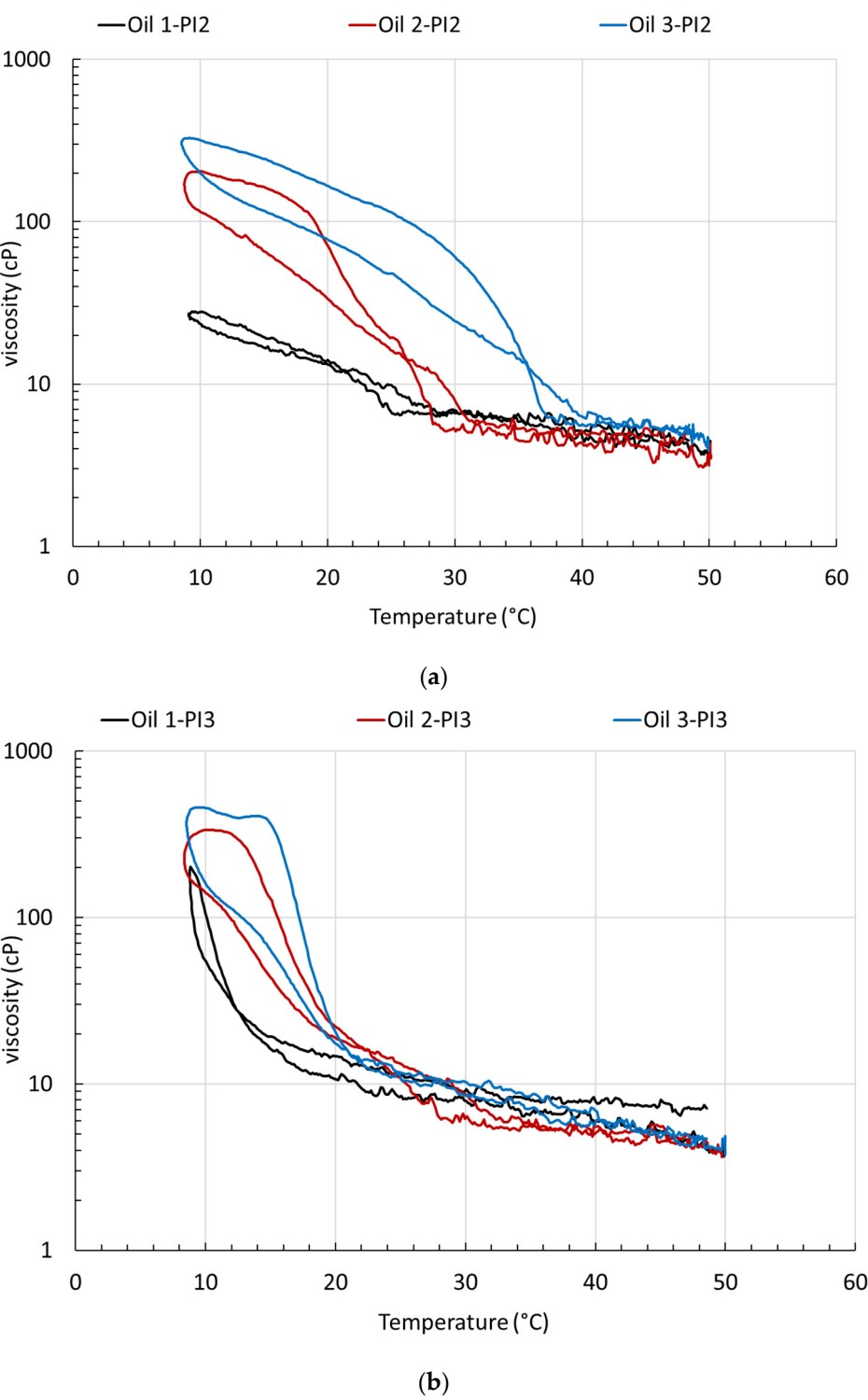

**Figure 5.** Viscosity as a function of the cooling/heating cycle 50 °C → 8 °C → 50 °C for oils (**a**) 1-PI2, 2-PI2, and 3-PI2 and (**b**) 1-PI3, 2-PI3, and 3-PI3, measured by applying a constant shear of $25\,s^{-1}$ and a cooling/heating rate of 1 °C/min.

Although the viscosities of oils 2-PI2 and 3-PI2 at the temperature at which the CPM images were taken as well as their overall profiles on cooling are quite different, the same cannot be said of oils 2-PI3 and 3-PI3. That the viscosities and profiles on cooling of the latter pair are quite similar yet their crystal morphologies are so different, is noteworthy. In other words, oils can exhibit similar viscosity profiles yet be governed by entirely different wax

crystal growth mechanisms. While the primary crystal morphologies of the oil pairs 2-PI2 and 2-PI3, containing composition *CW2*, and 3-PI2 and 3-PI3, containing composition *CW3*, are similar, secondary differences exist. A somewhat continuous network of smaller needle-like crystals is present in oil 3-PI2, that we have associated with higher viscosities and WATs in previous studies [12], yet absent in oil 3-PI3. Contrastingly, the crystal structures in oil 2-PI2 are less complex and extensive than those in oil 2-PI3. These opposing differences are likely responsible for the leveling of viscosities and profiles on cooling between oils 2-PI3 and 3-PI3 that are noted above.

The viscosity profiles of oils 3, 3-PI2 and 3-PI3, each containing wax composition *CW3*, show different efficacies for the two PIs with PI3 outperforming PI2 (Table 4). Whereas the oil treated with PI3 exhibits a notably lower viscosity profile above about 16 °C than that treated with PI2, a somewhat different trend emerges on inclusion of data pertaining to oils containing the shorter chain length wax compositions *CW1* and *CW2*. The viscosity profiles of oils 2-PI2 and 2-PI3, containing the shorter chain length wax composition *W2*, are lower overall than those of oils 3-PI2 and 3-PI3, although there are greater differences between the oils treated with PI2 than between those treated with PI3. Whereas the WAT of oil 2-PI2 is 30 °C, reduced from 37 °C for oil 3-PI2, that for oil 2-PI3, although more gradual, is somewhat unchanged relative to oil 3-PI3. In oil containing the shortest chain length composition *W1*, the profiles obtained for oils 1-PI2 and 1-PI3 are further reduced overall and quite similar to one another on cooling to 14 °C, at which point a clear WAT occurs in the latter oil but not the former. Clearly, greater differences between the profiles of oils treated with PI2 and PI3 are evident with increasing wax chain length.

Those oils treated with PI3 overlay much more closely than the analogous oils treated with PI2. These data demonstrate that while PI3 treats the different wax compositions somewhat equally well, PI2 is notably less effective in treating the higher chain length wax compositions. Oil 1-PI3 features what appears to be the most pronounced hysteresis effect on heating above its WAT, however, the resulting viscosity profile is only 1–2 cP greater than its cooling profile. Nevertheless, no such effect is evident for oil 1-PI2. Although PI2 boasts a small, longer length component of C24-26 to its side-chain length distribution relative to PI3, the data show that this feature does not eventuate in more effective treatment of the longer chain wax compositions *CW2* and *CW3*. While unexpected, this result has been reported in a dodecane based system [12]. That PI3, having a lower chain density, i.e., a larger volume between chains, is better able to accommodate wax of different chain lengths, particularly the longer lengths found in wax compositions *CW2* and *CW3*, may point to a steric effect associated with greater wax sequestering ability. At odds with previous observations in dodecane, an examination of the effect of different concentrations of the same wax compositions on the viscosity profiles and crystal morphologies of oils treated with each PI might further expand our understanding of paraffin control strategies.

### 3.3. Rheological Characterization and Microstructure of Oil Containing Wax and Asphaltenes

Dark clusters, which are presumed to be asphaltene-rich deposits, are evident in the CPM images of oils 2A and 3A (Figure 6) as interruptions in the wax crystal networks, which are somewhat denser than those in oils 2 and 3, not containing asphaltene. Only the afore-mentioned clusters are visible in the image of oil 1A (image not shown).

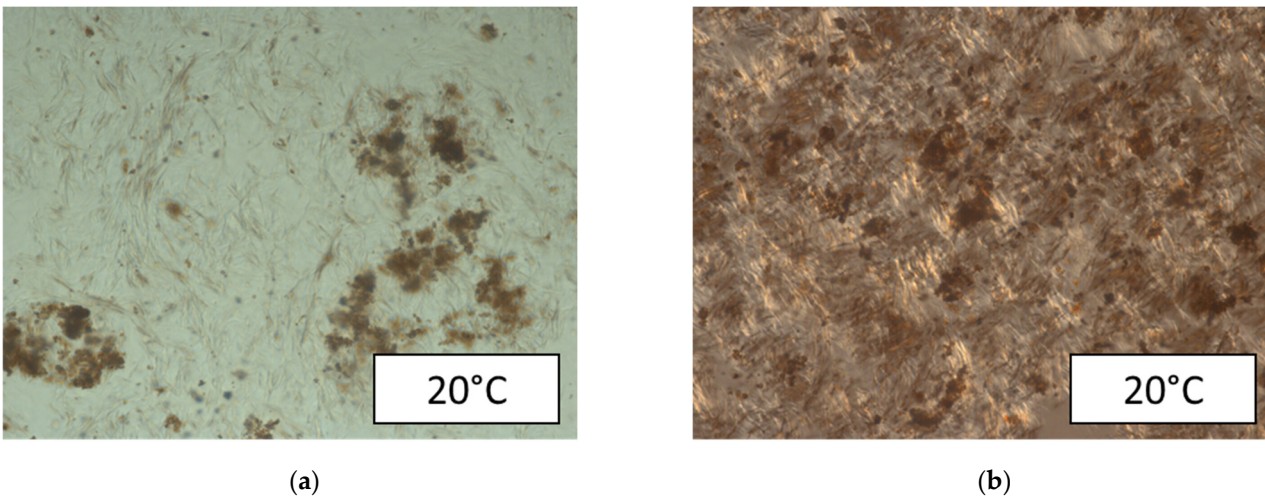

(**a**)                 (**b**)

**Figure 6.** CPM images at 20 °C of oils (**a**) 2A and (**b**) 3A. Wax crystals were not observed for oil 1A under the same conditions.

The presence of 1 wt. % asphaltenes in oil 1A provides a slight benefit by reducing the WAT from 27.5 to 25 °C and the maximum viscosity, reached on cooling to about 9 °C, from 834 to 655 cP (Figure 7, Table 4). However, slight increases in viscosity were observed in oils 2A and 3A, containing longer chain paraffin, relative to their asphaltene-free counterparts, oils 2 and 3. To a first approximation, however, no meaningful differences between the profiles are present.

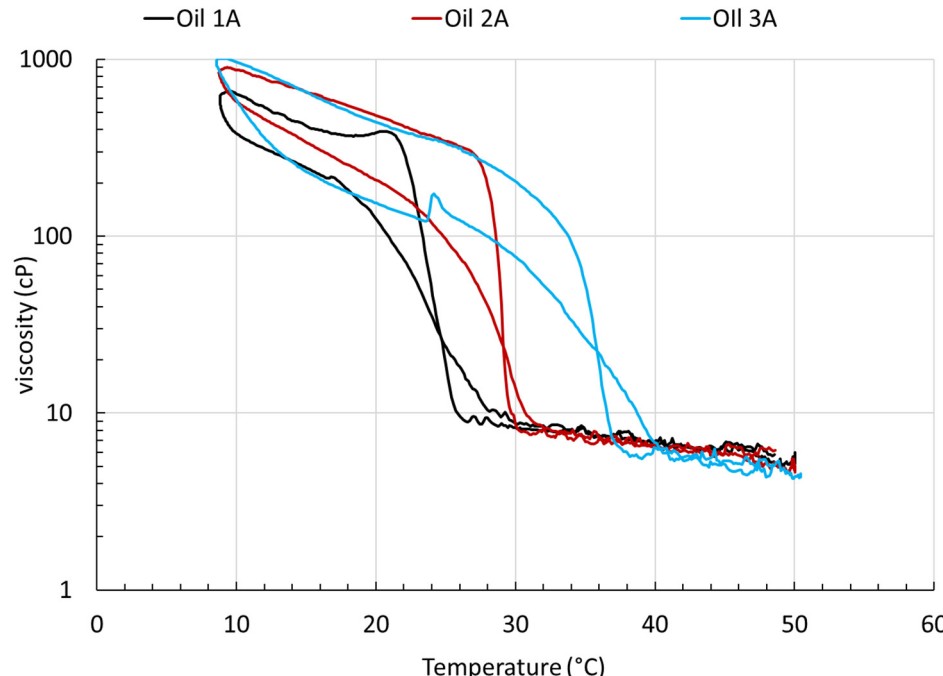

**Figure 7.** Viscosity as a function of the cooling/heating cycle 50 °C → 8 °C → 50 °C for oils 1A, 2A, and 3A, measured by applying a constant shear of 25 s$^{-1}$ and a cooling/heating rate of 1 °C/min.

## 4. Rheological Characterization and Microstructure of Oil Containing Wax and Asphaltenes, Treated with PI

Again, no crystal structures associated with wax could be observed by CPM for oils 1A-PI2 and 1A-PI3. Similar to the asphaltene-free oils, the oil pairs 2A-PI2 and 2A-PI3 (Figure 8a,c), each containing wax composition *CW2*, and 3A-PI2 and 3A-PI3 (Figure 8b,d),

each containing composition *CW3,* have similar crystal morphologies yet there are large differences between the oils in each pair. As such, primary differences in morphology again appear to be driven by the wax composition rather than the selection of PI in these cases, whereas the latter selection again eventuates in less prominent, secondary differences.

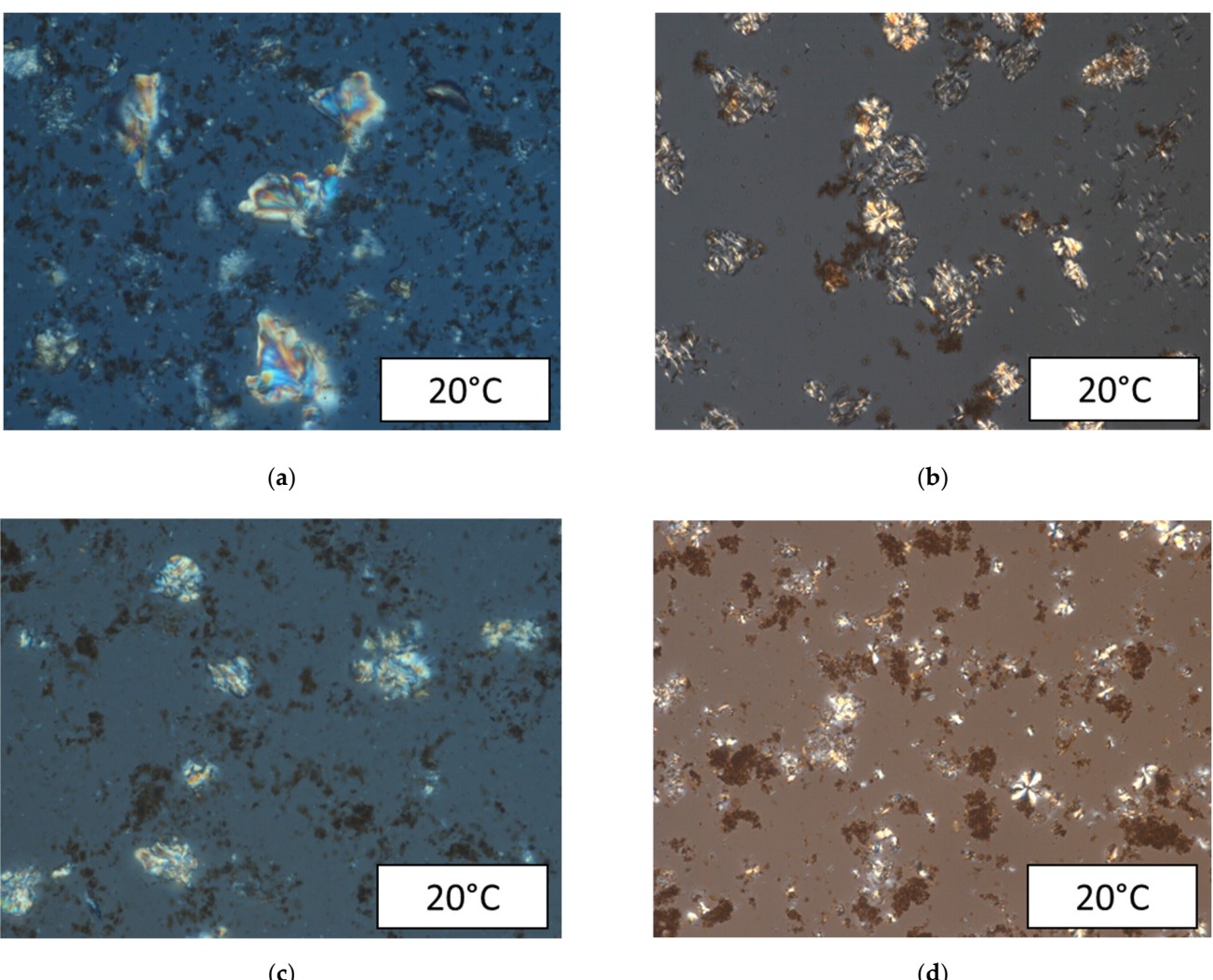

**Figure 8.** CPM images at 20 °C of oils (**a**) 2A-PI2 and (**b**) 3A-PI2, (**c**) 2A-PI3, and (**d**) 3A-PI3. Wax crystals were not observed for oil 1A-PI2 or 1A-PI3 under the same conditions.

Our previous studies have shown that synergetic effects between asphaltenes and PIs can considerably reduce the viscosity of oils containing both relative to those containing each separately [12]. The viscosities of oils 1A-PI2 and 3A-PI2 did not exceed 40 cP at any point during the cooling/heating process while that of oil 2A-PI2 reached a maximum of only 117 cP at 9 °C (Figure 9a). Of those oils treated with PI3, only oil 1A-PI3 exceeded this value, with a maximum of 202 cP at 9 °C (Figure 9b). No well-defined WATs were identifiable in any of the oils containing asphaltene, excepting oil 1A-PI3, for which a WAT of 16 °C is nearly identical to that of oil 1-PI3, not containing asphaltenes. Since the presence of asphaltenes does not appear to affect either the crystal morphologies of the oils (Figure 6 vs. 2) or their viscosity profiles (Figure 7 vs. 3) to a large degree, that the set of oils treated with PI2 and those treated with PI3 overlay more closely than their asphaltene-free counterparts is likely due to interactions involving the PI rather than those between the oil components themselves. Although the profiles are reduced overall, these data clearly demonstrate that differences in PI performance arising from differences in the chain length of the wax being treated are attenuated in the presence of asphaltenes, particularly in the

case of PI2, having the higher chain density. If the steric effect proposed earlier is at work, then interactions between PI and asphaltenes appear to increase wax sequestering ability to some extent.

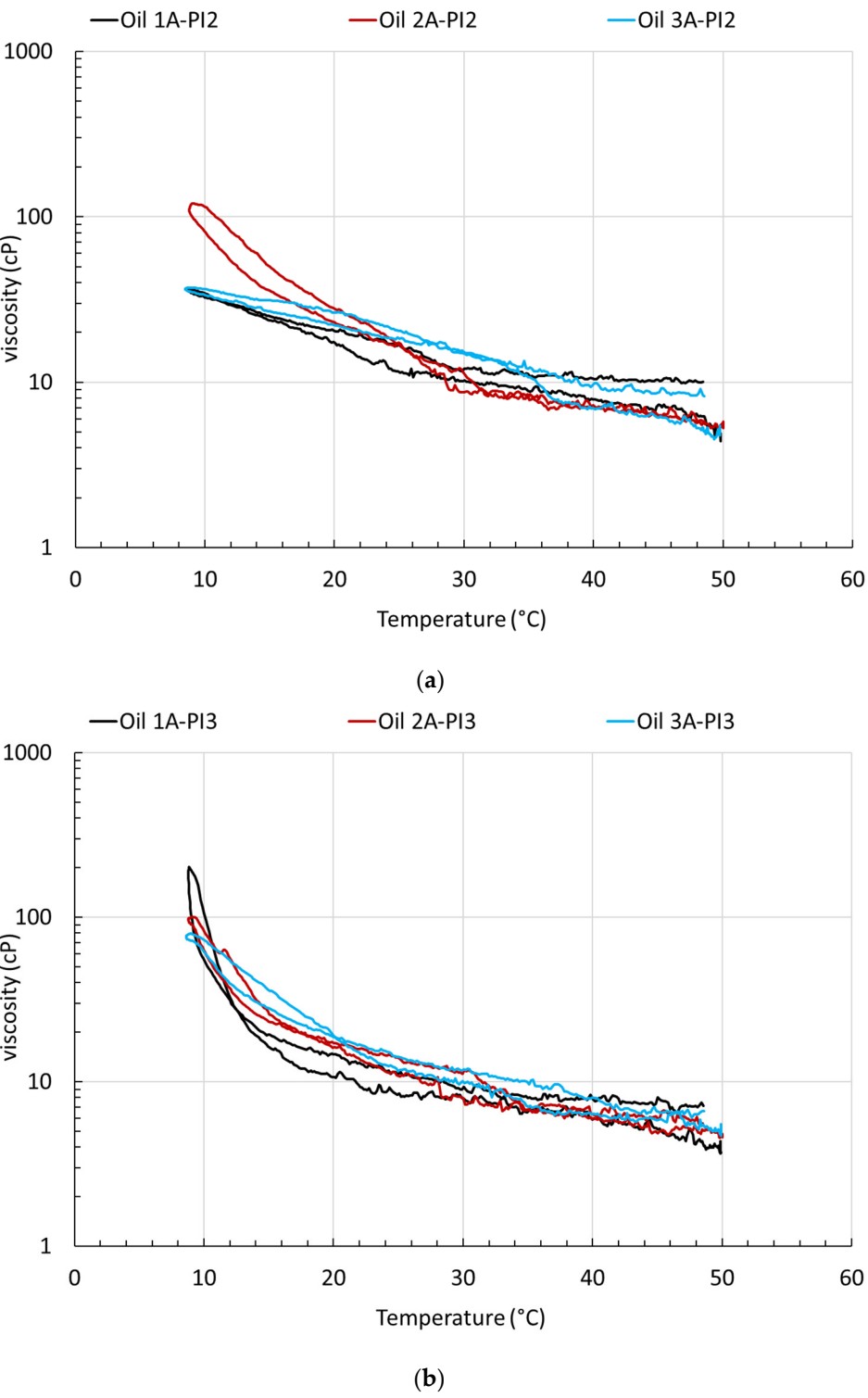

**Figure 9.** Viscosity as a function of the cooling/heating cycle 50 °C → 8 °C → 50 °C for oils (**a**) 1A-PI2, 2A-PI2, and 3A-PI2 and (**b**) 1A-PI3, 2A-PI3, and 3A-PI3, measured by applying a constant shear of 25 s$^{-1}$ and a cooling/heating rate of 1 °C/min.

In comparing the morphologies of oils 3-PI2 (Figure 4b) and 3A-PI2 (Figure 8b), which differ the greatest in their viscosity profiles, the wax crystal clusters present in oil 3A-PI2

appear less inter-connected than those in oil 3-PI2, with the smaller network of somewhat continuous crystals present in the latter being largely absent in former. Differing in their profiles by the second greatest extent, multi-clusters present in oil 3-PI3 give way to more discrete, albeit similar, clusters in oil 3A-PI3.

Treated with either PI, the viscosity profiles of oils containing wax compositions *CW2* and *CW3* are significantly improved in the presence of asphaltenes, however, the oils containing wax composition *CW1* are little changed. As such, the positive effects associated with asphaltenes are not realized in oil containing wax having the shortest chain lengths. These data support a steric origin given that the effect would logically be smaller, or even absent, in such oils. The viscosity of oil 2A-PI2 exceeds that of oils 1A-PI2 and 3A-PI2 below 18°C, however, it is perhaps more noteworthy that the viscosity profile of oil 2A-PI2 is quite similar to that of oil 2A-PI3, i.e., the performances of PI2 and PI3 in oil containing wax composition *W2* are similar. Interpreting the data from this perspective, the viscosity profiles of oils 1A-PI2 and 3A-PI2 are lower than their analog oils 1A-PI3 and 3A-PI3, treated with PI3. That PI2 is more effective in treating wax composition *W3* is opposite the trend observed in the absence of asphaltenes. It follows that the mechanism responsible for the superior performance of PI3 in the absence of asphaltenes, be it steric or otherwise, is largely negated in their presence, allowing other, secondary effects, to surface. The wider breadths of the chain length distributions of wax in compositions *CW1* and *CW3* relative to *W2* may account for these data. If so, at least in some cases, the breadth of the distribution may override average chain length, i.e., molecular weight [21,22]. The inclusion of waxes having similar average chain lengths additional PIs having systematic variations in chain density may shed light as to the nature of these effects.

## 5. Conclusions

Despite the wide use of wax inhibition chemicals in the oil and gas industry, factors affecting their mode of action and efficacy are not well understood due to the complexities of the oils being treated. In this work, WT crude was combined with wax consisting of varying average chain lengths and, to some extent, breadths of their chain length distributions. The performance of the PIs were compared with emphasis on differences arising from wax chain length with and without asphaltenes.

Primary differences in morphology appear to be driven by the wax composition, with secondary differences arising from the choice of PI. Oils with similar viscosity profiles are not always governed by similar morphologies. We therefore recognize that the measurement of viscosity over a cooling/heating cycle is but only one way to evaluate the effectiveness of a PI and the impact of wax composition and other components commonly found in crude oils. Future work aims to include cold finger testing to measure the amount and composition of wax deposition on cooling and DSC measurement to identify differences in the wax crystallization regime and the effect of PI on those regimes. The effectiveness of both PIs unsurprisingly follows the trend *CW1* > *CW2* > *CW3*, more effectively treating wax consisting of shorter rather than longer chains. The performance of PI2 is more greatly influenced by wax composition than that of PI3, with PI2, having a higher chain density, being less effective in treating wax of higher chain lengths than PI3. At odds with previous observations pertaining to dodecane-based oil, these trends suggest a steric effect in which PIs having lower chain densities are better able to accommodate wax of varying lengths. Because PIs of higher chain lengths likely boast higher paraffin sequestering ability, however, we recognize the need for experiments designed to probe the effect of wax concentration in addition to the effect of wax chain length.

Synergetic effects arising from the interaction of paraffin, asphaltenes and PI eventuate in lower viscosities than that for wax-containing oils containing either asphaltenes or PI alone. Differences in the performance of the PIs arising from differences in wax composition are diminished in the presence of asphaltenes, particularly in the case of PI2, owing to more effective treatment of wax consisting of longer chain lengths, with the treatment of shorter chains being largely unaffected. Combining this and previous observations,

we therefore propose that asphaltenes enhance electrostatic effects while lessening static effects. With the latter effect diminished, secondary effects present themselves, the nature of which requires additional research, possibly involving the breadth of the wax chain length distribution, to elucidate. Recognizing that the chemistry of the asphaltene, e.g., isoelectric point, undoubtedly plays an important role as well, experiments involving different asphaltenes are also needed.

**Author Contributions:** Conceptualization, O.M. and Q.P.N.; methodology, O.M. and Q.P.N.; validation, O.M., Q.P.N. and J.C.; writing—original draft preparation, O.M.; writing—review and editing, O.M., J.C., L.S., J.M.J. and Q.P.N.; supervision, Q.P.N.; project administration, L.S. and J.M.J.; funding acquisition, Q.P.N. All authors have read and agreed to the published version of the manuscript.

**Funding:** This research received no external funding.

**Institutional Review Board Statement:** Not applicable.

**Informed Consent Statement:** Not applicable.

**Data Availability Statement:** Not applicable.

**Conflicts of Interest:** The authors declare no conflict of interest. The funders had no role in the design of the study; in the collection, analyses, or interpretation of data; in the writing of the manuscript or in the decision to publish the results.

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
