# Peer review of "Impact of Paraffin Composition on the Interactions between Waxes, Asphaltenes, and Paraffin Inhibitors in a Light Crude Oil"

_colloids, doi:10.3390/colloids7010013_

Round 1

Reviewer 1 Report

This is a well-written and easy to follow piece of work. As presented, there are only a few corrections required, as listed below. However, included amongst the following list are also some specific technical points that I'd like the authors to address.

Line 27: paraffin inhibitors

Lines 41-42: ,,, sensitive to side-chain density rather than side-chain length, ,,,

Line 65: W1-W3 - and should these be italicized. Check on consistency of these definitions throughout (italics vs. bold, etc). See also lines 74, 78 and 79, for example

Line 109: concerning the choice of shear rate. The WAT effects seen at one shear rate may not appear, for example, at a higher shear rate, and could be greater at lower shear rates. Did the authors perform representative flow curve determinations (viscosity vs. shear rate) to determine that 25 s-1 is an appropriate shear rate to use for the present analysis? It would be instructive to include representative flow curves for each oil system, for instance. Please comment

Line 142: is disentanglement a better term?

Line 199: citation required for the previous work

Line 222: ditto

Line 237 (and earlier in Experimental): From where were the asphaltenes obtained? No indication has been given in the experimental section. How were the asphaltenes incorporated into the oil. It is known that asphaltenes are often difficult to (re-) solubilize into crude oil, and from the photomicrographs, it is apparent that the solid asphaltene particles have only been dispersed. This was also not explained in the authors' previously-cited paper (ref 12). 

Line 265: citation required to previous paper (presumably reference 12)

Line 297: ... their analog oils ...

Line 328: I might also suggest that DSC measurements could also help to identify crystallization differences between the different oils

Line 334: Would steric effects not be expected to be more pronounced in these non-aqueous environments?

Line 339: synergistic (line 339) or synergetic (line 265)? - consistency

Line 345: Alternatively, could one argue that the PIs adsorb preferentially to the solid asphaltene particles, and are therefore unavailable or at least less available to interact with the crystallizing waxes? This does not necessarily require electrostatic interactions to be invoked - if they are, perhaps the authors could expand on the discussion of the different modes of interaction.

Lines 386-387: Reference needs to be tidied up

Author Response

Dear Reviewer:

Please see attached the authors' responses to your comments. Thanks

Reviewer 2 Report

This paper reported an interesting study about the effectiveness of wax additives depending on the wax composition, the structure of the additive, and the presence of asphaltene. The samples were well designed to show the different effects, the involved experimental techniques appear adequate and sensitive for the subject of study, and the obtained results appear consistent and relevant.

The paper’s purpose fit well with the journal Colloids and Interfaces and consequently my recommendation is to ACCEPT the paper for publication.

However, I would list some points that should be considered as minor revisions in order to clarify the described work:

-      Author name “components” to w1, w2, w3 and “wax” to W1, W2, W3. I guess wi is a commercial wax too. The name “component” is confusing.

-      Author has to clarify if distributions in Fig 1, are experimental (GC or similar) or were computed from commercial information for wi.

-      Information in Table 2 should be better explained, the apparently little difference between PI2 and PI3 has to explain a big different behavior.

-      No information is given about the used crude oil (WT). At least a minimum characterization should be supplied (API, SARA, WAT).

-      Common study by CPM starts at high temperature and the cooling lead to solid wax formation. Authors have to explain why they use a different method.

-      In some cases, the information from both techniques appears inconsistent. For instance, 1 has a WAT of 27.5ºC but no solid phase is seen in CPM at 20ºC. Authors should comment.

-      Indicated WAT values apparently were obtained by viscosity measurements. It has to be clarified if some mathematical method was used because in some of the curves, it is not easy to obtain.

-      The viscosity vs. T curves are very relevant. In the description, two parameters appear relevant (WAT and the maximum viscosity). To offer a general view of all the experiments, I would suggest including a table with such values for all the systems studied.

-      CPM images show not only different morphology but also differences in size, which could be related to the different viscosity values. Authors could include such information.

-      Finally, nothing is said about reproducibility of the results. Authors should clarify how many experiments were carried out, or at least how they check the reproducibility, mainly for the viscosity determinations.

Author Response

(The authors gave the same response as above.)

Round 2

Reviewer 1 Report

Only one minor point: lines 129-130 should read "... (i.e. shear thinning) behavior could become ...

Otherwise the authors are thanked for their amendments.